# Safety Concerns Related to the Simultaneous Use of Prescription or Over-the-Counter Medications and Herbal Medicinal Products: Survey Results among Latvian Citizens

**DOI:** 10.3390/ijerph20166551

**Published:** 2023-08-09

**Authors:** Inga Sile, Renate Teterovska, Oskars Onzevs, Elita Ardava

**Affiliations:** 1Department of Applied Pharmacy, Riga Stradins University, 16 Dzirciema Street, LV-1007 Riga, Latvia; 2Latvian Institute of Organic Synthesis, 21 Aizkraukles Street, LV-1006 Riga, Latvia; 3Department of Pharmaceutical Chemistry, Riga Stradins University, 16 Dzirciema Street, LV-1007 Riga, Latvia; rsukele@gmail.com; 4Department of Pharmacy, Riga Stradins University Red Cross Medical College, 5 J. Asara Street, LV-1009 Riga, Latvia; elita.ardava@rcmc.lv; 5Department of Commerce, Turība University, 68 Graudu Street, LV-1058 Riga, Latvia; onzevs@turiba.lv

**Keywords:** adverse drug effects, consumer attitude, herb–drug interactions, herbal medicines, pharmacy

## Abstract

The use of herbal medicines is increasing worldwide. While the safety profile of many herbal medicines is promising, the data in the literature show important interactions with conventional drugs that can expose individual patients to high risk. The aim of this study was to investigate the experience of the use of herbal medicines and preparations and the risks of interactions between herbal and conventional medicines among Latvian citizens. Data were collected between 2019 and 2021 using a structured questionnaire designed for pharmacy customers in Latvia. Electronic databases such as Drugs.com, Medscape, and European Union herbal monographs were reviewed for the risk of drug interactions and potential side effects when herbal medicines were involved. The survey included 504 respondents. Of all the participants, 77.8% used herbal preparations. Most of the participants interviewed used herbal remedies based on the recommendation of the pharmacist or their own initiative. A total of 38.3% found the use of herbal remedies safe and harmless, while 57.3% of respondents regarded the combination of herbal and regular drugs as unsafe. The identified herbal medicines implicated in the potential risk of serious interactions were grapefruit, St. John’s wort, and valerian. As the risks of herb–drug interactions were identified among the respondents, in the future, both pharmacy customers and healthcare specialists should pay more attention to possible herb–drug interactions of over-the-counter and prescription medications.

## 1. Introduction

Botanicals find applications in a diverse array of products, including edibles, beverages such as teas and juices, as well as dietary supplements, herbal medicinal products, homeopathic products, cosmetics, etc. [1]. Over the last thirty years, there has been a significant increase in the consumption of herbal medicinal products and supplements [1,2]. As the worldwide consumption of herbal medicinal products keeps expanding, there is a growing acknowledgment of public health and safety concerns associated with them [3,4]. Many people use herbal medicines simultaneously with existing medications, believing that remedies of natural origin are safe. This could also be because most herbal medicines can be purchased worldwide without a prescription. Most herbal medicines are harmless in small doses. Plants have diverse chemical compositions and versatile pharmacological actions. They can interact with other medications and cause considerable health problems. Unfortunately, doctors and consumers do not always have the information to draw convincing conclusions about the interaction between herbal and prescribed medicines.

In many parts of the world, the risk of an interaction between medicines and herbal preparations increases every year. The concomitant use of both herbal preparations and medicines also increases the risk to patient health, leading to various side effects and risks of interactions [5,6]. Ineffective medical treatments increase the probability that the patient will enter the hospital during a disease flare-up. Researchers emphasize that various medications may have a negative effect on enzymes involved in drug metabolism [7]. Although considered natural, most herbal medicines can interact with other drugs, causing potentially harmful side effects or leading to loss of or decreased therapeutic benefits of the drugs. As a result, a person may have a false impression that drugs are ineffective or only slightly effective, thus negatively influencing the patient regarding the medication already prescribed. It should be noted that the information available from scientific sources and various international sources shows that additional attention should be given to the safety of different groups of medicinal products, with a focus on the groups at increased risk of drug interaction [2]. Taking into account the growing popularity of herbal medicines, there is an increased risk of clinical complications in groups of chronic disease patients [6,8]. Herbal products are the primary form of treatment, particularly among the elderly population who are also taking multiple prescription medications to manage their concurrent conditions. Consequently, this elevates the potential for adverse interactions between herbs, drugs, and diseases [6,9]. According to reports, not all risks associated with interactions between active herbal substances and conventional medications have been completely identified. Nevertheless, it is crucial to focus on those interactions for which reliable information is available [10].

The aim of this study was to investigate the experience of the use of herbal medicines and preparations and the risks of interactions between plants and conventional medicines among Latvian citizens. It is important to identify the attitudes of the respondents towards the safety of herbal medicines and products, and it is crucial to ascertain the respondents’ perspectives on whether they believe it is necessary to inform a pharmacist or physician about their use of herbal medicines and products while undergoing conventional drug treatment. The perception that naturally derived remedies are generally safe could lead to a reduced likelihood of patients disclosing their use of all herbal medicinal products to healthcare professionals. Hence, it becomes essential to have discussions about potential health hazards related to the simultaneous use of prescription and over-the-counter medications.

## 2. Materials and Methods

The methodology of this study was based on the completion of a structured questionnaire designed for visitors to community pharmacies who voluntarily agreed to participate. The survey included 504 consecutive respondents older than 18 years. Following the conventions of social studies, this methodology ensured a 3% margin of error with a confidence level of 5%. The survey was carried out from December 2019 to November 2021 in Latvian cities (Riga, Daugavpils, Ventspils, Liepaja, and Jelgava) and towns (Talsi and Ludza). The structured interviews using the questionnaire were carried out by pharmacy assistants from Rīga Stradiņš University, Red Cross Medical College. Participants had the opportunity to ask for advice concerning the questions included in the questionnaire if necessary. The questionnaire was approved by the Faculty of Pharmaceuticals of RSU Red Cross Medical College, Latvia (Appendix A).

The interviews were conducted in an isolated section in the sales area at community pharmacies. The questionnaire consisted of 4 parts and 15 mainly multiple-choice and Likert scale questions. Sociodemographic data such as age, sex, place of residence, and educational level of participants were collected in the first part. The second part was related to diseases. Information about the medicines used was collected in the third part. In the fourth part of the questionnaire, participants provided information on the medicinal plants they used. Respondents were requested to provide information regarding their experiences in utilizing herbs and drugs over the course of the previous year. 

The potential interactions between prescription, over-the-counter drug, and herbal medicinal product combinations were assessed using the Drugs.com Interaction Checker (https://www.drugs.com/ (accessed on 15 May 2023)), Medscape Drug Interaction Checker (https://www.medscape.com (accessed on 17 May 2023)), and herbal monographs of the European Union (EU) by the Committee on Herbal Medicinal Products (HMPC) and published by the European Medicines Agency (EMA) (https://www.ema.europa.eu/ (accessed on 20 May 2023)). Drugs.com offers a web-centric platform that provides drug-related information to medical professionals and patients. The interactive drug checker tool allows users to search for potential drug interactions, offering alerts concerning drug, alcohol, food, and disease interactions. The tool also provides information on the associated side effects and the degree of severity of each interaction. On the other hand, Medscape serves as an online clinical resource tool for healthcare professionals. In drug monographs, information on adverse effects, warnings, contraindications, and potential drug interactions is listed. Interaction alerts specify the severity level and suggest whether alternative therapy should be considered. European Union herbal monographs are a web-based supplier that includes information on interactions with other medicinal products and other forms of interaction. Drug interactions were categorized into three groups: serious, which is highly clinically important and require avoiding the combination or using an alternative drug; moderate, which has moderate clinical significance and should generally be avoided or may be used only in specific situations; and minor, which has minimal clinical significance, and their risks should be evaluated and minimized while considering alternative drugs. Since Drugs.com and Medscape are complementary, we carefully analyzed and compared any discrepancies between the databases, as well as the information provided by European Union herbal monographs. For instance, if one database indicated a moderate interaction while the other classified it as serious, we made a thoughtful decision to underscore the need for special attention by describing the interaction as serious. Through careful consideration and evaluation of the available evidence, our aim was to provide accurate and responsible information regarding the potential severity of the identified drug interactions.

This study was approved by the Red Cross Medical College of Riga Stradiņš University Ethics Committee (Approval number: 31; Appendix A). To ensure the anonymity of the participants involved in the questionnaire, the authors implemented rigorous data privacy measures. These included de-identification of data, using only aggregate data in reporting, ensuring the confidentiality of personal information at every stage, and obtaining informed consent from the participants. Additionally, any identifying information was securely stored and accessed only by the research team.

Although the data collection method used self-reported responses, potential errors associated with memory recall and social desirability bias were minimized by involving pharmacy assistants. These assistants assisted participants in remembering specific drug names by displaying the corresponding medication boxes available in the pharmacy.

With regard to the survey being conducted by pharmacy assistants, and due to the fact that their professional background and training may influence the data collected a standardized questionnaire was used to reduce potential bias and all pharmacy assistants received extensive training prior to the survey on how to accurately and objectively collect data. 

All data were verified by the leading researcher and double-checked by another experienced researcher/pharmacist.

Data were statistically evaluated using Microsoft Office Excel 2016 (Version 16.0; sourced from Microsoft Corporation, located in Redmond, Washington, USA). Descriptive statistics were used to summarize the characteristics of a data set. Measures of mean, central tendency, and frequency distribution describing the occurrence of data within the data set (count) were calculated.

## 3. Results

### 3.1. Demographics

During the survey period, a total of 504 respondents were interviewed. Of these respondents, 380 (75.4%) were female and 124 (24.6%) were male. The survey showed that the age distribution of the respondents was approximately evenly distributed. Most of them belonged to age groups of 40–49, 30–39, and 50–59 years. The smallest number of responders was in the age groups of 80–95 (*n* = 18, 3.6%). Most of the participants had secondary school (47.2%) and high school (46.8%) education, and a minority of responders obtained only basic education, mainly in the oldest age category. Out of all the respondents, there was only one individual who held a Ph.D. degree. More than half of the respondents (297; 58.9%) lived in the city (Table 1).

### 3.2. The Use of Herbal Medicines and Herbal Dietary Supplements and Their Recommended Sources

The respondents were queried about their daily usage of herbal preparations, which encompassed various forms such as teas, oils, tinctures, extracts, tablets, capsules, or syrups. Most respondents (*n* = 392, 77.8%) replied that they used herbal preparations as needed or during the disease course. Sixty-three (12.5%) respondents regularly used herbal preparations, while only 49 (9.7%) did not use herbal preparations daily. The majority of the participants (*n* = 239, 47.4%) took herbal medicines on the advice of a pharmacist, 217 (43.1%) took them on their own initiative, and 177 (35.1%) relied on the advice of their physician, while the recommendations of advertisements or a person without a medical background (for example, family, friends, and others) had less influence on their decisions (Figure 1).

### 3.3. Attitudes towards the Safety of Herbal Medicines

Out of the respondents, a significant portion (*n* = 193, 38.3%) perceived the use of herbal medicines as safe and without harm. However, 174 individuals (34.5%) expressed the belief that their usage was not entirely safe and could potentially lead to adverse effects, including allergies. More than a quarter of the respondents (*n* = 115, 22.8%) believed that the combination of herbal and conventional medicines was not safe and may affect the effects and efficacy of other medications. The remaining respondents (*n* = 22, 4.4%) reported that the safety of the medicines was not important to them.

### 3.4. Medicinal Plants Used among Individuals Who Used Herbal Products

Two-thirds of the respondents regularly used peppermint, coffee, chamomile, garlic, cranberry, lime tree flowers, green tea, onion, and ginger on a daily basis. The most commonly used plants are shown in Figure 2. In almost all age groups, the dominant herbal medicines were the same: peppermint, coffee, chamomile, garlic, and cranberry.

### 3.5. Respondents’ Health Concerns and Interest in Medication Side Effects

According to the survey findings, 46.2% of the respondents indicated that they were rarely or very rarely concerned about potential side effects when starting a new medicine, dietary supplement, or herbal preparation. A total of 4.4% of the respondents were never interested in side effects, while only 3 respondents were always interested (0.6%). A total of 48.8% of the respondents were often or very often interested in side effects.

During the survey, 56.2% of the respondents (*n* = 283) reported experiencing a long-term illness or chronic health condition within the past year. According to the data collected, musculoskeletal disorders, such as back pain, were common among the respondents (Table 2). High blood pressure, allergies, depression and anxiety, high blood cholesterol and fats, and gastric or peptic ulcer disease were also very common among respondents. Most respondents (*n* = 196, 38.9%) indicated that they had two chronic diseases, 27 (5.4%) indicated that they had three, and 24 (4.8%) had four chronic diseases. Three (0.6%) respondents indicated that they had experienced 7–8 chronic diseases or conditions in the last year.

Elderly patients exhibit a higher incidence of chronic diseases (R^2^ = 0.9426) and experience adverse effects more frequently (R^2^ = 0.6954). However, the correlation between the age of the user and the use of herbal medicine was not as strong. The determination coefficient (R^2^ = 0.5452) indicates that only 54.52% of cases for plant users were associated with age (Figure 3).

Most respondents (*n* = 217, 43.1%) reported that they had not had an acute disease in the last year. Most respondents had one (*n* = 138, 27.4%), two (*n* = 73, 14.5%), or three (*n* = 25, 5.0%) acute diseases. More than four acute diseases in the last year were reported by 16 (3.2%) respondents, and 35 (7.0%) respondents did not remember the exact number. Regarding acute conditions, almost half (48.4%) of the respondents used prescription drugs prescribed by a doctor.

A total of 29.4% of all respondents used four to ten prescription and over-the-counter drugs daily. A total of 16.1% of all respondents took three prescription and over-the-counter drugs, 22.2% took two, 16.5% took one, and 15.9% did not take any medication. Despite having a higher incidence of chronic disease (R^2^ = 0.9426) and using more active compounds simultaneously, as well as experiencing adverse effects more frequently, our study results indicate that the consumption of herbal products was not necessarily higher among elderly patients (Figure 4). 

A total of 70.0% of the respondents did not notice any side effects caused by the drugs or their interactions. Thirty percent of the respondents had experienced side effects of the medicines.

### 3.6. Potential Clinically Significant Interactions between Herbal Medicines and Conventional Drugs

The existence of interactions between herbal products and conventional drugs is evident, and some of these interactions have been well characterized and documented. In an examination of the potential interactions of the most common prescription and over-the-counter drugs used by the respondents with herbal medicines, the interactions of 66 active substances with 18 different plant species were identified (Table 3). The interactions may lead to clinically significant adverse reactions, such as serious, moderate, or minor reactions. The interactions of moderate significance have mostly been identified, which means that the combination of prescription and over-the-counter drugs with certain herbal medicines should usually be used with caution, avoided, or could be used only under special circumstances. Thirty-five combinations were classified as ‘serious’ and were characterized as highly clinically significant. This means that a combination should be avoided, or an alternative drug should be used. The herbal medicines implicated in the potential risk of serious interactions included grapefruit, St. John’s wort, and valerian. Most of the identified interactions involved potential alterations in the plasma concentration or effect of drugs, including nonsteroidal anti-inflammatory drugs (NSAIDs), diuretics, beta-blockers, HMG-CoA reductase inhibitors (statins), calcium channel blockers, paracetamol, and angiotensin-converting enzyme (ACE) inhibitors. Most herb–drug interactions are estimated to cause potential adverse outcomes associated with a higher risk of bleeding (or thrombosis), reduced efficacy, or changes in the bioavailability of the drug.

## 4. Discussion

In Latvia, the traditions of using plants to resolve various health problems are very old and are also widely described in folklore [47]. Most of the respondents in this study used medicinal plants in different dosages, and only a small number did not use herbal preparations daily. The regular utilization of medicinal plants, coupled with the observation that most participants perceive herbal medicines as safe, indicates a higher likelihood of adverse interactions between drugs and herbs. High levels of consumption of and self-medication with herbs have been reported in many other studies [2,48,49,50]. The respondents started using various herbal preparations mainly on the recommendation of a pharmacist. Although pharmacists, unlike doctors, do not have complete information on the medications used by a pharmacy visitor to warn about adverse effects in the use of herbal products and conventional drugs, these data show that the pharmacist plays an important role in correctly advising on the safe and effective use of medicines. Pharmacists are strongly advised to inquire directly with their patients about any herbal products they may be using prior to dispensing conventional drugs. A large number of respondents also reported that they began using herbal preparations on their own initiative. This situation can present the greatest risk. In other studies conducted in European countries as well as in Canada, a significant proportion of surveyed individuals used herbal remedies either independently or based on suggestions from non-medically trained individuals, without previous consultation with a health care professional [51,52,53,54]. The substantial rate of self-medication, coupled with the fact that users fail to inform their doctor or pharmacist about the herbal medicines they are using independently, represents a significant concern that has been extensively addressed in the literature but is not frequently discussed in practical settings. The fact that over-the-counter (OTC) medicines and food supplements are available without a prescription does not mean that they are safe to use, especially in combination with prescribed medication. Therefore, patients must be constantly advised by health care professionals. Due to the fact that the majority of respondents in Latvia use herbal products, several implications for the healthcare system could be discussed, including the need for better regulation and monitoring of herbal products, as well as improved education for both healthcare professionals and consumers regarding the safe use of these remedies. 

In Latvia, the general practitioner has access to the patient’s medical record, which contains information about their past use of prescription and OTC medications. This access to the medication record holds particular significance for elderly patients since side effects and potential drug interactions are frequently observed in this age group. A study by Qato et al. [55] reveals that approximately 50% of users engage in concurrent usage of over-the-counter (OTC) medications and dietary supplements. Among this group, it has been observed that roughly 2% of individuals exhibit potential major interactions between OTC medications and dietary supplements. González-Stuart [56] addressed the issue highlighting that older patients tend to withhold information from their healthcare providers regarding their use of herbal products. Consequently, this lack of disclosure limits the identification of potential interactions and health effects that may arise from the combination of conventional treatments and herbal products. It should be noted that in Latvia there is no information on the concomitant use of OTC and herbal products.

A large percentage of the respondents reported having at least two long-term diseases or chronic health conditions. Among them, back pain was frequently reported, which is usually treated with analgesics, as well as increased blood pressure, which is treated with multiple classes of antihypertensive medications such as beta-blockers, ACE inhibitors, or calcium channel blockers. When studying the literature on the interactions between herbal and conventional medicines, undesirable interactions between herbal medicines and analgesics and antihypertensive medications often appear [26,57]. Healthcare providers in Latvia should be mindful of potential interactions between herbal and conventional medicines used to treat back pain or hypertension. This awareness prompts them to inquire about patients’ use of herbal products, enabling better patient management and the development of individualized treatment plans. By acknowledging and addressing potential interactions, healthcare providers can minimize the associated risks and ensure optimal patient care.

Two-thirds of the Latvian respondents regularly used peppermint, coffee, chamomile, garlic, cranberry, lime tree flowers, green tea, onion, and ginger on a daily basis. Surprisingly, there was no notable variation in the reported usage of herbs across different age groups. When these data on the most commonly used plants are compared with those of other European countries, there are many differences. Some countries also report garlic, cranberries, chamomile, and lime tree flowers as the most commonly used plants; however, other often used plants include ginkgo, valerian, evening primrose, aloe, and artichoke [1,37,51,52].

The use of herbal medicine is influenced by various factors, and its correlation with incidence does not solely depend on the frequency of plant use. Notably, in the present study, the groups reporting adverse effects did not align with those who used herbal medicine the most. The utilization of plants may be influenced not only by age but also by cultural, social, and economic factors. To gain deeper insights into these aspects, future studies could incorporate more in-depth interviews for analysis.

Herbal medicine, like any form of medical treatment, can have both benefits and risks. While herbal remedies have been used for centuries and are often considered natural and safe, they can also pose significant complications, particularly when used improperly or without appropriate guidance. Some of the main complications associated with herbal medicine include toxicity, adverse reactions, and potential drug interactions, leading to reduced effectiveness or dangerous side effects when combined with medications. Allergic reactions, misidentification, and contamination of herbal products are also concerns, along with delayed treatment or misdiagnosis when relying solely on herbs [2,58,59]. The lack of scientific evidence further adds to the risks. It is important to note that herbal products are not risk-free, and the risk of drug interactions has not been sufficiently studied, so more research is needed in this field, and healthcare professionals should advise patients to take precautions. In general, patients should consult with a healthcare provider before taking any herbal medicine. People’s knowledge of the use of plants and their preparations is incomplete and can cause health risks; therefore, in the present study, the authors were able to assess whether there were undesirable risks between over-the-counter and prescription and herbal medicines used by the respondents.

Although there were mainly minor or moderate interactions between the most commonly used herbal medicines and the conventional drugs reported by the respondents, this indicates a potential risk of adverse events. Although the majority of interactions may only have minor clinical implications, certain interactions could present a serious threat to public health. It is very important to pay attention to the rarer cases identified in the survey, which have much more serious consequences, such as the use of St. John’s wort, valerian, and grapefruit. These cases should definitely be reported by patients to healthcare professionals. Herbal medicines like St. John’s wort can significantly impact the enzymatic activity of liver cytochromes, potentially affecting the efficacy of prescription medicines and leading to treatment failure. Moreover, when St. John’s wort is combined with anti-depressants, it may trigger serotonin syndrome, an adverse drug reaction that can be life-threatening [4,9]. Regarding grapefruit, even though grapefruit juice is not consumed for medicinal purposes, its components can inhibit the CYP3A4 enzyme in the small intestine. This inhibition leads to elevated blood levels of CYP3A4 substrate drugs, such as calcium channel blockers, statins, antidepressants, and estrogen. The increased drug levels can potentially result in dangerous hypotension, myopathy, liver toxicity, or even an increased risk of breast cancer. Hence, the interaction between grapefruit juice and certain medications can be risky and should be taken seriously [9,60]. In contrast, valerian does not affect several CYP isoenzymes, including CYP3A4, CYP2D6, CYP2E1, and CYP1A2 [61]. Valerian root preparations are widely used as a sleep aid, and concomitant use could enhance the effect of CNS depressants [62].

Concerning the limitations and scope of this study, it is worth noting that although the sample size was adequate and represented the population in Latvia, the number of participants in each age group could have been more similar. Due to the lack of access to the respondents’ history of medication use, and the information on the use of drugs and herbal products was provided solely by the respondents themselves, there is a possibility that the accuracy and reliability of the data may be affected. The results could be influenced by recall bias, social desirability bias, and subjective interpretations of respondents. However, during the survey, pharmacist assistants were available to explain any unclear aspects to respondents or help clarify the correct names of the drugs. Additionally, it is important to acknowledge that the study was conducted solely within the boundaries of Latvia, and the findings may not be directly applicable to other populations or regions with different healthcare systems or cultural contexts. In addition, it is worth noting the potential impact of COVID-19 on the results since data collection occurred during the pandemic. One might question whether it affected the frequency and manner of natural product use among Latvian citizens. However, it is not pertinent to discuss the influence of the COVID-19 pandemic in this particular manuscript. This is primarily due to the fact that the bulk of the data was gathered in 2020, a time when no COVID-19-related restrictions were in place in Latvia. Furthermore, respondents were asked to report on medications and herbs used in the previous year, which refers to 2019, a period prior to the pandemic.

The purpose of the present study was to show the current situation and raise awareness of this subject in Latvian society and the existing risks of concomitant use of herbal and conventional medicines among pharmacists and physicians, thus protecting the health of consumers. Patients may use herbal medicines concurrently with conventional drugs, leading to the possibility of potentially serious adverse events. It is the responsibility of healthcare professionals to stay well-informed about the increasing clinical evidence of interactions between herbs and drugs. By being knowledgeable about these interactions, healthcare professionals can better assess and manage the risks associated with the combined use of herbal medicines and conventional drugs, ensuring the safety and well-being of their patients.

## 5. Conclusions

Survey respondents who took herbal medicines, especially those containing garlic, grapefruit, St. John’s wort, valerian, and others that affected the pharmacokinetic and pharmacodynamic properties of over-the-counter and prescribed medications, were at risk of experiencing different degrees of herb–drug interactions. Therefore, patients should always be advised by a healthcare professional. Considering the current pharmaceutical policy in Latvia, enhancing the quality of guidance can be achieved by granting both physicians and pharmacists access to the general medical record, which contains the history of a patient’s usage of prescription and over-the-counter drugs. Having this comprehensive information readily available will enable healthcare professionals to make more informed decisions, provide better care, and effectively address potential interactions or adverse effects that may arise from the combined use of herbal medicines and conventional drugs. Healthcare professionals should also be educated on where to look for information on possible drug interactions.

## Figures and Tables

**Figure 1 ijerph-20-06551-f001:**
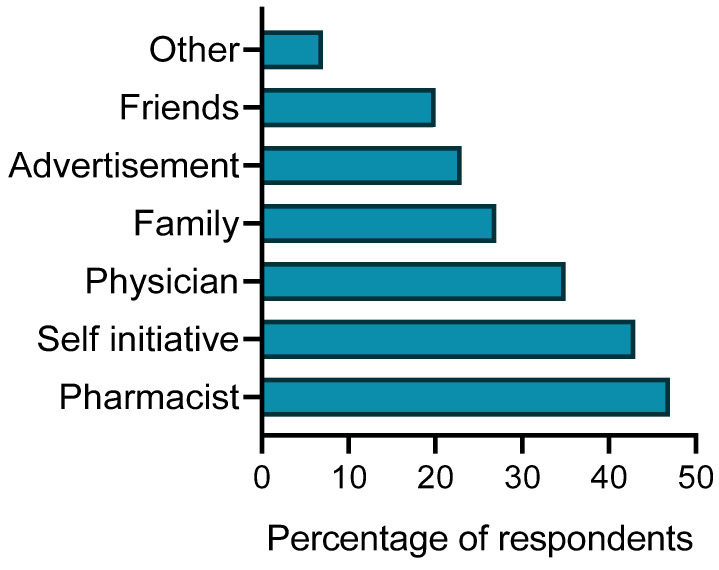
Recommendation sources for the use of herbal medicines purchased by respondents (%).

**Figure 2 ijerph-20-06551-f002:**
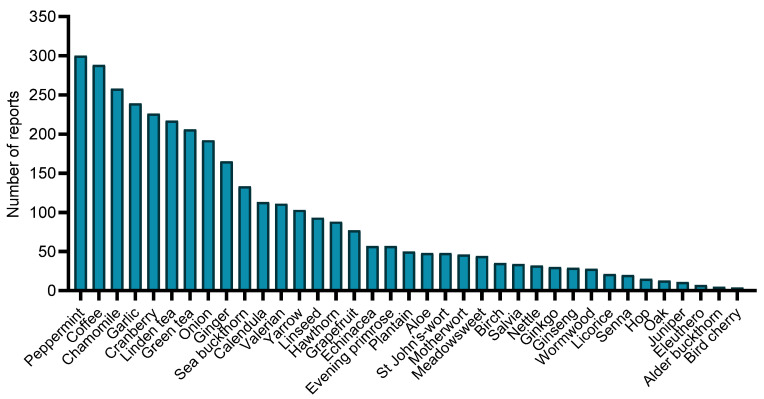
The most frequently mentioned plants by respondents.

**Figure 3 ijerph-20-06551-f003:**
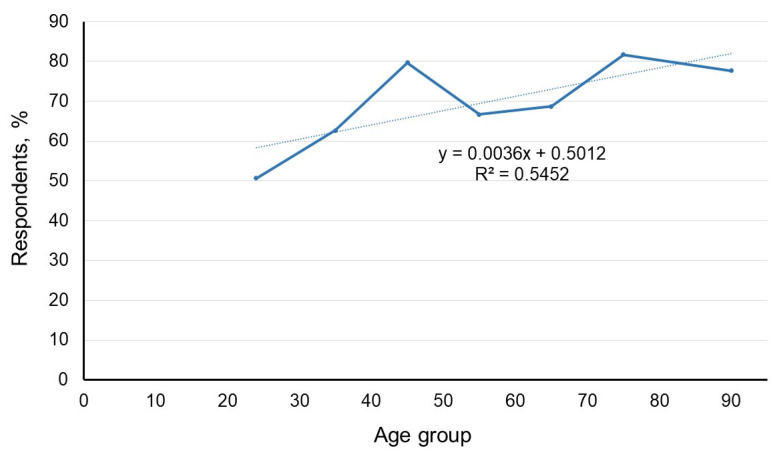
Respondents using herbal medicine by age group.

**Figure 4 ijerph-20-06551-f004:**
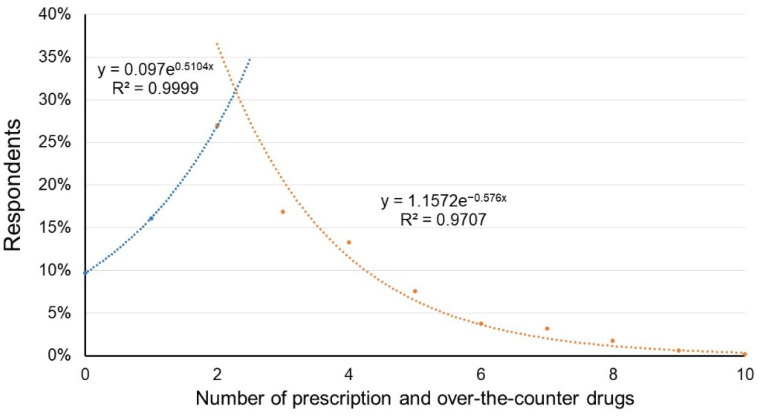
The number of substances used (prescription and over-the-counter drugs) by age median. The blue line represents young people aged 18 to 39 years, while the orange line represents older patients.

**Table 1 ijerph-20-06551-t001:** Demographic characteristics of the respondents.

Characteristics		Frequency	Percentage, *n* = 504
Age	18–29	65	12.9
	30–39	103	20.4
	40–49	106	21.0
	50–59	91	18.1
	60–69	70	13.9
	70–79	51	10.1
	80–95	18	3.6
Sex	Female	380	75.4
	Male	124	24.6
Education	Basic education	29	5.8
	Secondary education	238	47.2
	Higher education	236	46.8
	Doctoral degree	1	0.2
Region	City	297	58.9
	Town	119	23.6
	Countryside	88	17.5

**Table 2 ijerph-20-06551-t002:** Long-term conditions or chronic diseases.

Long-Term Health Condition	Frequency (%)
Back pain	185 (69.5%)
High blood pressure	157 (59.0%)
Allergies	72 (27.1%)
Depression, anxiety	56 (21.1%)
High blood cholesterol and fats	56 (21.1%)
Gastric or peptic ulcer disease	55 (20.7%)
Heart disease	49 (18.4%)
Kidney and urinary tract diseases	49 (18.4%)
Different types of arthritis	41 (15.4%)
Lung diseases	29 (10.9%)
Anaemia or any other blood condition	17 (6.4%)
Osteoporosis (bone atrophy)	15 (5.6%)
Diabetes	15 (5.6%)
Cancer	12 (4.5%)
Liver diseases	5 (2.3%)
Other (thyroid problems, bone fractures, joint injuries, gynaecologic diseases, venous problems, and others)	62 (23.3%)

**Table 3 ijerph-20-06551-t003:** Potential clinically significant interactions between herbal medicines and the most common conventional drugs used by respondents.

Herbal Medicine	Conventional Drug	Clinical Result of Interaction	Classification of Drug Interaction *	Comments	References
Grapefruit *Citrus paradisi*	Dextromethorphan	Grapefruit increases the level or effect of dextromethorphan and ivabradine by affecting hepatic/intestinal enzyme CYP3A4 metabolism.	Serious		[11]
St. John’s wort *Hypericum perforatum*	Amitriptyline, citalopram, clomipramine, dextromethorphan, escitalopram, mirtazapine, paroxetine, sertraline, trazodone, venlafaxine	Coadministration may potentiate the risk of serotonin syndrome, which is a rare but serious and potentially fatal condition thought to result from hyperstimulation of brainstem 5-HT1A and 2A receptors.	Serious		[12,13,14]
	Alprazolam, atorvastatin, carbamazepine, clopidogrel, diazepam, felodipine	St. John’s wort decreases the level or effect of the drug by affecting hepatic/intestinal enzyme CYP3A4 metabolism.	Serious	Special attention should be given when using any drug that is metabolized by CYP3A4, CYP2C9, CYP2C19, or P-glycoprotein in combination with another substance because it may lead to a reduction of plasma concentrations.	[15,16]
	Dabigatran	Coadministration with potent inducers of P-glycoprotein may significantly reduce the bioavailability of dabigatran following oral administration.	Serious		[17,18]
	Drospirenone, oestradiol, ethinylestradiol, ivabradine, loratadine, nifedipine, prednisolone, sildenafil, rivaroxaban, trazodone	St. John’s wort decreases the level or effect of these drugs by affecting hepatic/intestinal enzyme CYP3A4 metabolism.	Serious	Coadministration with St. John’s wort may reduce the efficacy of contraceptive hormones. Nonhormonal contraception is recommended.	[17,19]
Valerian *Valeriana officinalis*	Alprazolam, clonazepam, codeine, diazepam, gabapentin, phenobarbital	Coadministration of valerian and these drugs increases sedation effects.	Serious		[20,21]
Alder buckthorn *Frangula alnus*	Hydrochlorothiazide, indapamide	Prolonged or excessive use of laxatives can enhance the pharmacological impact of diuretics. Laxatives have the potential to lead to substantial depletion of fluids and essential electrolytes like sodium, potassium (resulting in hypokalemia), magnesium, and zinc. These effects could be cumulative when combined with the effects of diuretics.	Moderate		[22]
Echinacea *Echinacea purpurea*	Acetaminophen	Coadministration with echinacea may increase the plasma concentrations and the risk of adverse effects of drugs that are substrates of CYP450 1A2.	Moderate		[23,24]
	Amlodipine, atorvastatin	Coadministration with echinacea may alter the plasma concentrations and therapeutic effects of drugs that are substrates of CYP450 3A4.	Moderate		[23,24]
Garlic *Allium sativum*	Aceclofenac, aspirin, diclofenac, ibuprofen	Garlic has the potential to enhance the effects of anticoagulants, platelet inhibitors, and thrombolytic agents, which may lead to an increased risk of bleeding.	Moderate	Garlic powder, aged garlic preparations, garlic oil, and fresh garlic all contain antiplatelet activity. Some isolated reports have linked chronic and high dietary intake of garlic to bleeding complications.	[25,26]
Ginger *Zingiber officinale*	Aceclofenac, aspirin, diclofenac, ibuprofen	Ginger has the potential to enhance the effects of anticoagulants, platelet inhibitors, and thrombolytic agents, which could potentially increase the risk of bleeding.	Moderate		[27,28]
Ginkgo *Ginkgo biloba*	Aspirin, ibuprofen, diclofenac	Ginkgo may potentiate the risk of bleeding associated with anticoagulants, platelet inhibitors, nonsteroidal anti-inflammatory drugs (NSAIDs), and thrombolytic agents.	Moderate	Consumption of ginkgo should be avoided during the use of coagulation-modifying agents and at least two weeks prior to surgery.	[20,29]
	Hydrochlorothiazide, indapamide	Blood pressure may increase following the addition of ginkgo while receiving a thiazide diuretic.	Moderate		[25]
Ginseng *Panax ginseng*	Aceclofenac, aspirin, diclofenac, ibuprofen	Ginseng may potentiate the effects of anticoagulants, platelet inhibitors, and thrombolytic agents, possibly increasing the risk of bleeding.	Moderate		[30]
Grapefruit *Citrus paradisi*	Atorvastatin	Grapefruit juice can increase atorvastatin blood levels.	Moderate	The combination of certain medications, such as atorvastatin, with grapefruit juice can increase the risk of side effects like liver damage and a rare but serious condition known as rhabdomyolysis, which involves the breakdown of skeletal muscle tissue. Therefore, it is recommended to limit grapefruit juice consumption to no more than 1 quart per day while undergoing atorvastatin treatment to mitigate these risks.	[31,32]
Green tea *Camellia sinensis*	Aspirin, diclofenac, ibuprofen	Combination may increase the risk of bleeding.	Moderate		[33]
Licorice *Glycyrrhiza glabra*	Amlodipine, bisoprolol, metoprolol, nebivolol, perindopril, ramipril	Licorice use has been associated with hypertension and may antagonize the effects of antihypertensive agents.	Moderate	The use of licorice should be avoided in conjunction with diuretics, cardiac glycosides, corticosteroids, stimulant laxatives, or other medications that may worsen electrolyte imbalance.	[20,34]
	Indapamide, hydrochlorothiazide	Chronic use of licorice may potentiate the hypokalemic effects of some diuretics and other drugs that deplete potassium.	Moderate	Severe hypokalaemia can lead to muscle paralysis, rhabdomyolysis, metabolic alkalosis, cardiac arrhythmias, and respiratory arrest.	[20]
Motherwort *Leonurus cardiaca*	Phenylephrine	Motherwort decreases both blood pressure and heart rate, while phenylephrine increases both blood pressure and heart rate.	Moderate		[35,36]
Nettle *Urtica dioica; U. urens*	Aceclofenac, aspirin, diclofenac, ibuprofen	Nettle reduces the anticoagulant effect of aceclofenac, aspirin, diclofenac, and ibuprofen and lower plasma concentrations.	Moderate		[5]
Senna *Cassia senna*; *C. angustifolia*	Hydrochlorothiazide, indapamide	Prolonged or excessive use of laxatives can enhance the pharmacological impact of diuretics. Laxatives have the potential to cause considerable loss of fluids and essential electrolytes, such as sodium, potassium (resulting in hypokalemia), magnesium, and zinc. These effects may add up to the effects of diuretics.	Moderate		[37]
Siberian ginseng *Eleutherococcus senticosus*	Aceclofenac, aspirin, diclofenac, ibuprofen	Coadministration increases anticoagulation.	Moderate		[38]
	Atorvastatin	St. John’s wort decreases the level or effect of atorvastatin by P-glycoprotein (MDR1) efflux transporter.	Moderate		[39]
	Amlodipine, omeprazole	Coadministration with potent inducers of CYP450 2C19 and 3A4 may significantly decrease the plasma concentrations of the drug.	Moderate	The interaction is likely to also occur with esomeprazole, an enantiomer of omeprazole.	[40]
Birch *Betula*	Hydrochlorothiazide, indapamide	Birch increases the effects of these drugs by pharmacodynamic synergism.	Minor		[41]
Chamomile *Matricaria chamomilla*	Aspirin	In theory, consuming large amounts of chamomile could increase the risk of bleeding in patients undergoing treatment with drugs that impact the body’s ability to control bleeding, such as anticoagulants, platelet inhibitors, thrombin inhibitors, thrombolytic agents, or medications that often lead to low platelet counts (thrombocytopenia).	Minor	For patients who have extensively used chamomile before starting anticoagulation, antiplatelet, or thrombolytic therapy, it is crucial to consider the potential for interactions.	[20,42]
Evening primrose *Oenothera biennis*	Aspirin	Theoretically, the use of evening primrose oil with anticoagulants or antiplatelet aggregation drugs may increase the risk of bleeding in some patients.	Minor		[42]
Garlic *Allium sativum*	Amlodipine, bisoprolol, hydrochlorothiazide, indapamide, metoprolol, nebivolol, perindopril, ramipril	Garlic has been found in some studies to lower blood pressure and may theoretically potentiate the effects of hypotensive agents. There have been no reports of clinical hypotension associated with the concomitant use of garlic and antihypertensive agents.	Minor		[43]
Grapefruit *Citrus paradisi*	Amlodipine	Grapefruit increases the level or effect of amlodipine by affecting hepatic/intestinal enzyme CYP3A4 metabolism.	Minor		[11]
Green tea *Camellia sinensis*	Acetaminophen	Combination may increase the risk of bleeding.	Minor		[44]
	Atorvastatin, rosuvastatin	Coadministration with green tea may decrease oral bioavailability.	Minor		[45]
Sage *Salvia officinalis*	Phenylephrine	Sage lowers blood pressure, while phenylephrine increases blood pressure	Minor		[35,46]

* Drug interactions were categorized into three groups: serious, which are highly clinically important and require avoiding the combination or using an alternative drug; moderate, which has moderate clinical significance and should generally be avoided, or may be used only in specific situations; and minor, which has minimal clinical significance, and their risks should be evaluated and minimized while considering alternative drugs.

## Data Availability

Data are available upon request.

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
