# Peer review of "Safety Concerns Related to the Simultaneous Use of Prescription or Over-the-Counter Medications and Herbal Medicinal Products: Survey Results among Latvian Citizens"

_ijerph, 2023, doi:10.3390/ijerph20166551_

Round 1

Reviewer 1 Report

Author selected an interesting topic for potential risk of herbal medicine interactions with prescribed and/or over the counter medications. Authors evaluated potential drug-herbal interactions according to several resources drugs.com, EMA, Medscape, PDR herbal, and European Union herbal monographs. Other evidence based resources like natural medicine database were not used in the study. The study is descriptive and can be improved. I have some questions as followed.

1.     Do authors include 504 consecutive patients or use a predefined sample size? Is so, Please clarify the sample size formula

2.     If possible, please classify potential drug interactions by level of evidence

3.     Please bring the incidence of herbal drug interactions in elderly versus other patients

4.     4. can authors bring factors may influence the incidence of herbal drug interactions (correlation analysis), for example, did underlying conditions (back pain,..) influenced herbal medications use or not?

5.     In the first table (demographic data), its better to bring other data including underlying disease, having or not having acute disease or not for better understanding

6.     In the table 3, please bring the evidence of interactions. For example the interactions between lycor and antihypertensive agents

a.     Or interactions between echinacea and and beclomethasone is not serious in naturalmedicine database. Please reconsider if possible

7.     Its better to reclassify the table 3, first the major, then moderate and minor herbal-drug interactions

8.     In the discussion section, please bring major complication of herbal medicine

Reviewer 2 Report

In my opinion, this is an interesting and valuable study that deserves to be published as long as the following modifications are made:

Title and Introduction

- According to the authors, data collection was conducted from December 2019 to November 2021. Therefore, it is evident that the majority of the data collection phase took place during the COVID-19 pandemic, although this was likely not the original intention of the authors. It is crucial to acknowledge this aspect in the introduction (and in other sections of the manuscript) as it is apparent that the occurrence of the pandemic could have significantly influenced the results in terms of the frequency and manner of natural product use among the citizens of Latvia.

- Therefore, I leave it to the authors' discretion to consider changing the title of the manuscript to something like: "Safety Concerns Related to the Concurrent Use of Medications and Herbal Medicinal Products: Survey Results Among Latvian Citizens During the First Year of the COVID-19 Pandemic," if they deem it appropriate.

- Likewise, it is essential to cite the following reference both in the introduction and discussion of the manuscript: "Nino-Orrego MJ, Baracaldo-Santamaría D, Patricia Ortiz C, Zuluaga HP, Cruz-Becerra SA, Soler F, Pérez-Acosta AM, Delgado DR, Calderon-Ospina CA. Prescription for COVID-19 by non-medical professionals during the pandemic in Colombia: a cross-sectional study. Ther Adv Drug Saf. 2022 May 24;13:20420986221101964. doi: 10.1177/20420986221101964. PMID: 35646306; PMCID: PMC9136451," along with other relevant studies that should be sought by the authors.

Methods

- In order to promote open science policy, I strongly recommend that the authors share the questionnaire that was used as an annex to the study.

- In the methodology, the authors describe using different sources of information for searching interactions, such as drugs.com and Medscape. It is clear that these systems are complementary since they do not always yield the same results. What criteria (and what was the procedure) did the authors employ in case of finding discrepancies between the databases?

- Upon reviewing the results, many of the responses appear to correspond to past situations. What was the timeframe during which the use of herbal medicinal products was assessed?

- It is important to mention the recall bias (or memory bias) in this study, along with other potential biases that are characteristic of survey-based observational studies. Likewise, it is crucial to mention the strategies employed by the authors to control for biases in the study.

- I do not see the corresponding section on ethical aspects, so it is unclear whether the project was approved by a research ethics committee or not. Additionally, I suggest including the informed consent form mentioned by the authors as an annex to the study. Lastly, it is important to mention the mechanisms employed by the authors to protect the identity of participating individuals.

Results

- In line with the previously mentioned points, I suggest clearly stating the date when the state of emergency was instituted in Latvia and separating the results before and after the pandemic, as it may lead to interesting findings.

- Section 3.6 is very interesting as it corresponds to the potential interactions between natural products and the medications commonly consumed by patients. However, since these are potential (theoretical) interactions without a real evaluation of clinical consequences (e.g., alteration in plasma levels, decompensation in disease control, over-anticoagulation, etc.), I suggest changing the section title and Table 3 to "Potential Drugs-Herbs Interactions."

- In the sentence "Most of the identified interactions involved potential alterations in the concentration…," I suggest adding the word "plasma" before "concentration."

- In the sentence "Most herb-drug interactions are estimated to cause unsafe outcomes associated with the risk of bleeding, reduced efficacy, or bioavailability of the drug," I suggest revising it to "Most herb-drug interactions are estimated to cause potential adverse outcomes associated with a higher risk of bleeding (or thrombosis), reduced efficacy, or changes in the bioavailability of the drug."

- In Table 3, the statement "Motherwort increases and phenylephrine decreases sedation effects" seems to be incomplete. Please review it. 

- The same observation applies to the phrases "These drugs increase and nettle decreases anticoagulation effects" and "Sage increases and phenylephrine decreases sedation effects." It is necessary to specify exactly what is being increased. Are they plasma levels? Of which drug(s)?

- Clinical results and comments in Table 3 need references. The same applies to Table 4.

- I do not understand the reason for separating Tables 3 and 4 when they could be presented in one table, as they seem to correspond to the same outcome of the study.

- In the Results section, when it is stated, "Most respondents (196) indicated that they had two chronic diseases, 27 indicated that they had three, and 24 had four chronic diseases. Three respondents indicated that they had experienced 7-8 chronic diseases or conditions in the last year," I suggest adding the percentages for greater clarity for readers. The same observation applies to the following paragraph: "Most respondents (n=217) reported that they had not had an acute disease in the last year. Most respondents had one (n=138), two (n=73), or three (n=25) acute diseases. More than four acute diseases in the last year were reported by 16 respondents, and 35 respondents did not remember the exact number."

- Regarding Table 2, is it possible to correlate the underlying diseases of each subject with the natural products they were taking? This is important as it would allow establishing the frequency of "off-label" use for natural products, as well as possible interactions between the herbal remedy and the disease.

- This sentence is confusing: "A total of 186 respondents, or 60.9%, did not notice any side effects caused by the drugs or their interactions. Thirty percent of the respondents had experienced side effects of the medicines, while 9.1% did not remember having any side effects." It is unlikely for a person to have experienced adverse effects and not remember them, so I recommend combining the values of 60.9% and 9.1% into one. Regarding the fact that 30% of the subjects experienced at least one adverse effect, it is crucial to include a table describing the side effects and the implicated herbal product, considering that this is an article on pharmacovigilance.

Discussion

- Overall, I perceive that the Discussion is very poor and lacks depth, as in many cases, it merely repeats the results without delving much into the significance and implications of the findings.

- In the sentence "In other studies conducted in European counties as well as in America," there is a typo with the word "counties," which should actually be "countries." Additionally, it is not appropriate to refer to America as a whole when only one of the studies cited at the end of the statement included subjects from Canada, but not from Central or South America.

- In the statement "However, it should be noted that there is no information on the concomitant use of over-the-counter and herbal products," the context is not clear. Is it referring to Latvia or in general? At least one reference is needed to support this statement.

- This paragraph "A large percentage of the respondents reported having at least two long-term diseases or chronic health conditions. Among them, back pain was frequently reported, which is usually treated with analgesics, as well as increased blood pressure, which is treated with multiple classes of antihypertensive medications. When studying the literature on the interactions between herbal and conventional medicines, undesirable interactions between herbal medicines and analgesics and antihypertensive medications often appear [19,20]" is practically a repetition of what has already been presented in the Results section without additional elements that contribute to the Discussion.

- When it is stated: "People's knowledge of the use of plants and their preparations is incomplete and can cause health risks; therefore, in the present study, the authors were able to assess whether there were undesirable risks between over-the-counter and prescription and herbal medicines used by the respondents," it is important to clarify if any type of pharmaceutical intervention was carried out to correct these risky behaviors.

- As mentioned earlier, it is important to establish that the majority of the study was conducted during the first year of the COVID-19 pandemic. Therefore, it is necessary to contrast the results obtained before and after the declaration of the state of emergency in Latvia, as well as compare the results of this study with other similar studies conducted during the pandemic.

- The Discussion lacks a paragraph of self-criticism regarding the limitations and scope of the present study.

Round 2

Reviewer 2 Report

I appreciate the authors for providing responses and accommodating the majority of my recommendations. I believe the manuscript has significantly improved and is practically ready for acceptance for publication. I also understand the reasoning presented for not changing the manuscript's title or alluding to the COVID-19 pandemic, but as some readers may have the same concern I had, I suggest providing clarity on that point in the discussion with the arguments presented in response 1 of the title and the introduction. Similarly, I think it is worth including response 2 from the Methods section as it may be valuable information for the readers.
